# Exogenous Serotonin (5-HT) Promotes Mesocotyl and Coleoptile Elongation in Maize Seedlings under Deep-Seeding Stress through Enhancing Auxin Accumulation and Inhibiting Lignin Formation

**DOI:** 10.3390/ijms242317061

**Published:** 2023-12-02

**Authors:** Xiaoqiang Zhao, Jiayao Li, Yining Niu, Zakir Hossain, Xiquan Gao, Xiaodong Bai, Taotao Mao, Guoxiang Qi, Fuqiang He

**Affiliations:** 1State Key Laboratory of Aridland Crop Science, College of Agronomy, Gansu Agricultural University, Lanzhou 730070, China; zhaoxq3324@163.com (X.Z.); lijiayao1018@163.com (J.L.); bxd15293898130@163.com (X.B.); m15294392789@163.com (T.M.); qigx1321@163.com (G.Q.); hefq6125@163.com (F.H.); 2State Key Laboratory for Crop Genetics and Germplasm Enhancement, Nanjing Agricultural University, Nanjing 210095, China; 3Swift Current Research and Development Centre, Agriculture and Agri-Food Canada, Swift Current, SK S9H 3X2, Canada; zakir.hossain@agr.gc.ca

**Keywords:** maize, deep-seeding stress, mesocotyl/coleoptile elongation, auxin accumulation, lignin formation, serotonin (5-HT), gene expression

## Abstract

Serotonin (5-HT), an indoleamine compound, has been known to mediate many physiological responses of plants under environmental stress. The deep-seeding (≥20 cm) of maize seeds is an important cultivation strategy to ensure seedling emergence and survival under drought stress. However, the role of 5-HT in maize deep-seeding tolerance remains unexplored. Understanding the mechanisms and evaluating the optimal concentration of 5-HT in alleviating deep-seeding stress could benefit maize production. In this study, two maize inbred lines were treated with or without 5-HT at both sowing depths of 20 cm and 3 cm, respectively. The effects of different concentrations of 5-HT on the growth phenotypes, physiological metabolism, and gene expression of two maize inbred lines were examined at the sowing depths of 20 cm and 3 cm. Compared to the normal seedling depth of 3 cm, the elongation of the mesocotyl (average elongation 3.70 cm) and coleoptile (average elongation 0.58 cm), secretion of indole-3-acetic acid (IAA; average increased 3.73 and 0.63 ng g^−1^ FW), and hydrogen peroxide (H_2_O_2_; average increased 1.95 and 0.63 μM g^−1^ FW) in the mesocotyl and coleoptile were increased under 20 cm stress, with a concomitant decrease in lignin synthesis (average decreased 0.48 and 0.53 A_280_ g^−1^). Under 20 cm deep-seeding stress, the addition of 5-HT activated the expression of multiple genes of IAA biosynthesis and signal transduction, including *Zm00001d049601*, *Zm00001d039346*, *Zm00001d026530*, and *Zm00001d049659*, and it also stimulated IAA production in both the mesocotyl and coleoptile of maize seedlings. On the contrary, 5-HT suppressed the expression of genes for lignin biosynthesis (*Zm00001d016471*, *Zm00001d005998*, *Zm00001d032152*, and *Zm00001d053554*) and retarded the accumulation of H_2_O_2_ and lignin, resulting in the elongation of the mesocotyl and coleoptile of maize seedlings. A comprehensive evaluation analysis showed that the optimum concentration of 5-HT in relieving deep-seeding stress was 2.5 mg/L for both inbred lines, and 5-HT therefore could improve the seedling emergence rate and alleviate deep-seeding stress in maize seedlings. These findings could provide a novel strategy for improving maize deep-seeding tolerance, thus enhancing yield potential under drought and water stress.

## 1. Introduction

Maize (*Zea mays* L.) is one of the three major cereal crops in the world; it serves as food, animal feed, and an industrial material. During seed germination, maize is often exposed to drought stress causing growth retardation and significant yield losses worldwide [1,2,3,4]. The normal maize sowing depth is 3~5 cm; because of the abundant evaporation and moisture loss in the soil surface, maize cannot absorb enough water to initiate seedling emergence in shallow sowing [1]. In agricultural practice, maize deep seeding (≥20 cm) is an important cultivation strategy to ensure seedlings’ emergence and survival under drought stress [5,6,7]. So far, a limited number of maize varieties with strong deep-seeding tolerance have been developed and widely used in (semi-) arid areas. For example, “P1213733 (Komona)”, an elite maize variety, was widely cultivated at a 30 cm depth or more in western Mexico and southwestern United States [8]; maize variety “40107” was planted to a sowing depth of 26 cm in (semi-) arid areas of China [1,9].

However, deep seeding decreased the seed emergence rate [10]. Plants use different organs to push the young plumule to the soil surface. For example, wheat (*Triticum aestivum* L.) [11,12] and barley (*Hordeum vulgare* L.) [13,14] seedlings come up to the soil surface through the organs’ elongation between the coleoptile and the first internode. The elongation and growth of the mesocotyl played an important role in promoting seedling emergence in rice (*Oryza sativa* L.) [15,16]. Our previous studies showed that maize seedling emergence occurs mainly through the cooperative elongation of the mesocotyl [1] and coleoptile [7]. In addition, hydrogen peroxide (H_2_O_2_) will be accumulated in the germ while it comes out from the soil surface and has exposure to light, which inhibits the production of PAO (polyamine oxidase) inhibitor 2-HEH (2-hydroxyethyl hydrazine), resulting in a large amount of lignin accumulation and the subsequent hardening of the mesocotyl cell wall and slowing down of the mesocotyl growth [17,18,19,20,21]. There is increasing evidence that the elongation of the mesocotyl is affected by exogenous hormones, such as gibberellin A3 (GA_3_) [22,23], indole-acetic acid (IAA) [24], salicylic acid (SA) [25], abscisic acid (ABA) [26], zeatin (ZT) [27], cytokinin (CTK) [28], ethylene (ET) [29], strigolactone (SL) [28], and brassinosteroid (BR) [30].

Indole is a compound of pyrrole and benzene with the chemical formula C_8_H_7_N, also known as benzopyrrole. Indole compounds widely exist in nature; it is often used as a growth regulator in plants [31]. The natural plant growth regulator indole 3-acetic acid, IAA and indole derivatives, mediates a large number of growth and developmental processes, such as cell elongation, cell division, cell differentiation, and morphogenesis. IAA is a typical indole hormone, which can promote the elongation and growth of maize mesocotyl cells during deep sowing [32]. Kutschera and Wang [33] found that exogenous IAA significantly extended the mesocotyl of maize in vitro, and cell elongation was mainly achieved through acidification and relaxation of the mesocotyl cell wall. Serotonin (5-HT), also known as serotonin, is an indole derivative with the molecular formula C_10_H_12_N_2_O. In addition to its functions in plant growth and development, flowering, fruit ripening, and seed formation [34], 5-HT also plays an important role in plant stress tolerance [35,36]. For example, Tian et al. [37] showed that exogenous 5-HT eliminated excessive H_2_O_2_ by increasing the antioxidant enzyme activity in rape seedlings, alleviated membrane lipid peroxidation, and enhanced the content of osmotic regulatory substances to alleviate drought stress of rape seedlings. However, the effect of 5-HT on the cooperative elongation of the mesocotyl and coleoptile in maize under deep-seeding stress has not been reported. Considering that both 5-HT and IAA are indole derivatives synthesized from tryptophan [38], it is reasonable to propose that they likely share a common regulatory mechanism regarding the role in regulating mesocotyl elongation. Therefore, in this study, the effect of different concentrations of 5-HT on mesocotyl and coleoptile elongation under deep-seeding stress was evaluated using two maize inbred lines. The changes in growth indexes and physiological metabolism, as well as gene expression levels involved in IAA biosynthesis, auxin signal transduction, and lignin biosynthesis, were studied to elucidate the mechanism of the 5-HT-mediated mitigation of deep-seeding stress. This study will provide valuable information for alleviating maize deep-seeding stress and improving yield, as well as maize drought tolerance breeding.

## 2. Results

### 2.1. Maize Seedling Phenotypes Treated with or without 5-HT under Deep-Seeding Stress

Growth and development of the seedlings of two maize inbred lines were significantly (*p* < 0.05) inhibited by deep seeding at 20 cm for 10 days (Figure 1A–C). Compared with a 3 cm seeding depth control treatment (CK(+)), the seedling emergence rate, seedling weight, and seedling length under the 20 cm deep-seeding stress treatment (CK(-)) were significantly reduced. The seedling emergence rate of the two maize inbred lines was reduced to 0%, while the seedling weight was reduced by 57.7% and 25.8%, and the seedling length was reduced by 50.9% and 27.0%, respectively (Figure 1A–C). These data showed that 20 cm deep seeding drastically affected the emergence rate and significantly inhibited the morphogenesis of maize seedlings.

Intriguingly, the growth inhibition of maize seedlings under 20 cm deep-seeding stress was significantly relieved by exogenous application of 5-HT (*p* < 0.05) (Figure 1A–C). Compared with the CK(-) treatment, different concentrations of 5-HT under 20 cm deep-seeding stress have different degrees of mitigation for seedling phenotypes. The growth indexes of the three traits were most significantly improved by the 20 cm seeding depth + 2.5 mg/L 5-HT application (T4) treatment. The seedling emergence rate of ly119-2 and ZRX87-1 was 95% and 50%, respectively; their seedling weight was 2.42 and 2.03 times that of the CK(-) treatment, respectively; and their seedling length was 2.70 and 1.86 times that of the CK(-) treatment (Figure 1A–C). Secondly, under the 20 cm seeding depth + 2.0 mg/L 5-HT application (T3) treatment, the seedling emergence rate of both inbred lines reached 60% and 15%, respectively; their seedling weight was 2.11 and 2.02 times that of the CK(-) treatment, respectively; and their seedling length was 2.21 and 1.73 times that of CK(-) treatment (Figure 1A–C). However, in the 20 cm seeding depth + 1.0 mg/L 5-HT application (T1), 20 cm seeding depth + 1.5 mg/L 5-HT application (T2), and 20 cm seeding depth + 3.0 mg/L 5-HT application (T5) treatments, the gain in the three growth parameters of both materials was relatively small; the seedling emergence rates were 0%, 5%, and 30% (ly119-2) and 0%, 0%, and 0% (ZRX87-1), respectively; the seedling weight was 1.23, 1.18, and 2.24 times (ly119-2) and 1.11, 1.26, and 0.93 times (ZRX87-1) that of the CK(-) treatment, respectively; the seedling length was 1.62, 1.28, and 2.47 times (ly119-2) and 1.19, 1.31, and 1.19 times (ZRX87-1) that of CK(-) treatment (Figure 1A–C). The finding confirmed that 5-HT significantly alleviated deep-seeding stress in seedlings.

### 2.2. Effect of Exogenous 5-HT on Maize Mesocotyl and Coleoptile Growth under Deep-Seeding Stress

The mesocotyl and coleoptile of maize were significantly extended synchronously in the deep soil layer [1,3]. Compared with the CK(+) treatment, the mesocotyl length, coleoptile length, mesocotyl and coleoptile total length, and mesocotyl-to-coleoptile ratio of the two inbred lines were significantly increased under the CK(-) treatment (*p* < 0.05). The increases were 4.92, 1.48, 2.80, and 3.33 times (ly119-2) and 1.77, 1.05, 1.40, and 1.68 times (ZRX87-1) that of the CK(+) treatment, respectively (Figure 2A,B,G,H). Under the CK(-) treatment, the mesocotyl weight of inbred line ly119-2 increased by 71.3%, while it decreased by 10.9% in ZRX87-1; the treatments had no impact on the coleoptile weight of the inbred lines (*p* < 0.05) (Figure 2E,F). The CK(-) treatment significantly reduced the mesocotyl coarse and coleoptile coarse of the inbred lines by 56.1%, 15.6% (ly119-2), and 31.2%, 25.7% (ZRX87-1), respectively (Figure 2C,D). These data revealed the increase in seedlings’ mesocotyl and coleoptile length and the reduction in coarse growth during deep seeding.

Compared with the CK(-) treatment, the mesocotyl length, coleoptile length, mesocotyl coarse, coleoptile coarse, mesocotyl weight, coleoptile weight, and mesocotyl and coleoptile total length of these inbred lines were increased first in response to 5-HT application at a 20 cm sowing depth and then decreased with increased concentration (Figure 2A–G). The mesocotyl-to-coleoptile ratio showed a downward trend (Figure 2H). For ly119-2 seedlings, the T4 treatment showed the greatest increase in mesocotyl length (89.1%), coleoptile length (77.4%), mesocotyl weight (47.3%), and mesocotyl and coleoptile total length (66.3%); the T5 treatment showed the greatest increase in coleoptile coarse (166.4%) and coleoptile weight (119.9%); and the T3 treatment showed the greatest increase in mesocotyl coarse (280.9%) relative to the CK(-) treatment (Figure 2A–G). Similarly, for ZRX87-1 seedlings, the T4 treatment displayed the greatest increase in mesocotyl length (34.4%), coleoptile length (86.8%), mesocotyl weight (89.7%), coleoptile weight (46.5%), and mesocotyl and coleoptile total length (54.3%); the T1 treatment displayed the greatest increase in mesocotyl coarse (9.7%); and the T2 treatment displayed the greatest increase in coleoptile coarse (36.4%) relative to the CK(-) treatment (Figure 2A–G). These results revealed that the addition of 5-HT in deep seeding promoted both the longitudinal and lateral growth of the maize mesocotyl and coleoptile to ensure seedling emergence from the deep soil layer.

### 2.3. Stimulative Effects of Exogenous 5-HT and Deep-Seeding Stress on Physiological Responses of Maize Mesocotyl and Coleoptile

When subjected to deep-seeding treatment, the mesocotyl and coleoptile of the seedlings of two maize inbred lines were stressed and a series of physiological and biochemical metabolisms were disturbed. The IAA and hydrogen peroxide (H_2_O_2_) contents of the mesocotyl and coleoptile were significantly increased in the CK(-) treatment compared with the CK(+) treatment (*p* < 0.05). Specifically, the IAA contents of the mesocotyl and coleoptile were 2.81 and 1.55 times that of the CK(+) treatment in ly119-2 and 1.15 and 1.11 times that of the CK(+) treatment in ZRX87-1, respectively. The H_2_O_2_ contents of the mesocotyl and coleoptile were 2.81 and 1.83 times that of the CK(+) treatment in ly119-2, while they were 4.21 and 2.16 times that of the CK(+) treatment in ZRX87-1, respectively (Figure 3A–D). In addition, the lignin content in the mesocotyl and coleoptile decreased significantly (*p* < 0.05) in the CK(-) treatment, with values of 21.78% and 25.84% in ly119-2 and 32.77% and 4.15% in ZRX87-1, respectively, of that in the CK(+) treatment (Figure 3E,F).

Furthermore, the IAA content in the mesocotyl and coleoptile of the inbred lines was first increased and then decreased with the increase in the 5-HT concentration under 20 cm deep-seeding stress (Figure 3A,B). However, the H_2_O_2_ and lignin levels showed an opposite trend; they decreased first and then increased (Figure 3C,D). The IAA content in the mesocotyl and coleoptile of the T4 treatment was increased most significantly (*p* < 0.05) compared with the CK(-) treatment, which were 2.64 and 1.38 times that of the CK(-) treatment in ly119-2 and 1.36 and 1.30 times that of the CK(-) treatment in ZRX87-1, respectively (Figure 3A,B). The H_2_O_2_ and lignin contents due to the 5-HT stimulation under deep-seeding stress were decreased to different degrees (*p* < 0.05) (Figure 3C–F). The H_2_O_2_ content in the mesocotyl was decreased by 12.49% (T1), 13.87% (T2), 32.20% (T3), 67.49% (T4), and 6.33% (T5) in ly119-2 and 17.22% (T1), 28.79% (T2), 40.51% (T3), 61.14% (T4), and 7.64% (T5) in ZRX87-1, respectively, compared with the CK(-) treatment. The H_2_O_2_ content in the coleoptile was decreased by 12.28% (T1), 20.32% (T2), 48.10% (T3), 37.13% (T4), and 25.44% (T5) in ly119-2 and 3.72% (T1), 17.87% (T2), 48.22% (T3), 55.42% (T4), and 3.23% (T5) in ZRX87-1, respectively (Figure 3C,D). The lignin content in both the mesocotyl and coleoptile of ly119-2 were significantly decreased (*p* < 0.05) compared with the CK(-) treatment, which decreased by 16.48% (T1), 13.07% (T2), 32.95% (T3), 56.82% (T4), and 4.55% (T5) in the mesocotyl and by 19.10% (T1), 26.13% (T2), 24.62% (T3), 46.23% (T4), and 20.10% (T5) in the coleoptile, respectively. The lignin content of ZRX87-1 was significantly decreased (*P* < 0.05) compared with the CK(-) treatment, which decreased by 9.85% (T1), 3.03% (T2), 25.76% (T3), 49.24% (T4), and 14.39% (T5) in the mesocotyl and decreased by 6.49% (T1), 12.97% (T2), 49.19% (T3), 57.30% (T4), and 7.03% (T5) in the coleoptile, respectively (Figure 3E,F). These data revealed that exogenous 5-HT enhances auxin accumulation and inhibits lignin production, resulting in the promotion of mesocotyl and coleoptile elongation in maize under deep-seeding stress.

### 2.4. Relationships among All Traits under Deep-Seeding Stress

To better understand the physiological complexity of deep-seeding stress on maize, we constructed a framework of relations among all 17 traits tested, based on between-group linkage cluster analysis, Pearson pairwise correlation analysis, and principal component analysis (PCA). The between-group linkage cluster analysis showed that when the Euclidean distance was 5.2, the 17 tested traits were classified into three unrooted groups (Figure 4A). The A cluster only included the seedling emergence rate, which represented the overall seedling emergence; the B cluster included 11 traits and represented the cooperative elongation of both the mesocotyl and coleoptile and their regulatory factors; and the C cluster included five traits and represented seedling morphological development. Furthermore, Pearson pairwise correlation analysis revealed 98 groups with a significant (*p* < 0.05) correlation between two traits (Figure 4B). These findings suggested that the inherent relationships among multiple traits were complex in maize treated with or without 5-HT at the 20 cm and 3 cm sowing depth, which further contributed to information overlapping. Thus, it was appropriate to conduct PCA, based on the eigenvalues and cumulative variances for all traits, for which the first four principal components (PCs; PC1, PC2, PC3, and PC4) accounted for 90.735% of the total variance, of which the eigenvalues were larger than 1.0 (Figure 4C). These PCs were the further linear combination of the different traits based on their variable loadings (Figure 4D). Specifically, the mesocotyl length, coleoptile length, mesocotyl and coleoptile total length, mesocotyl weight, coleoptile weight, IAA content in the mesocotyl and coleoptile, and lignin content in the mesocotyl and coleoptile were the primary traits in PC1, which accounted for 49.330% of the total variance. The mesocotyl-to-coleoptile ratio and H_2_O_2_ content in the mesocotyl and coleoptile were the primary traits in PC2, which accounted for 24.793% of the total variance. Seedling length was the primary trait in PC3, which accounted for 9.030% of the total variance. The seedling emergence rate and coleoptile coarse were the primary traits in PC4, which accounted for 7.583% of the total variance.

### 2.5. Comprehensive Mitigation Effect of 5-HT under Deep-Seeding Stress

To comprehensively evaluate the mitigation effect of different 5-HT applications on deep-seeding tolerance, the 17 tested traits of both maize inbred lines under all treatments were used for the qualitative identification of deep-seeding tolerance via between-group linkage cluster analysis. The results showed that two maize lines under seven treatments were obviously divided into three groups (Figure 5A). ZRX87-1 and ly119-1 cultured under the CK(+) treatment were the A group, suggesting that the two lines at the 3 cm normal sowing depth had best seedling emergence and establishment; ZRX87-1 and ly119-1 cultured under the T2, T1, and CK(-) treatments as well as ZRX87-1 cultured under the T5 treatment were the C group, showing that the corresponding 5-HT concentrations in these treatments did not significantly alleviate maize resistance to deep-seeding stress; ZRX87-1 and ly119-1 cultured under the T3 and T4 treatments as well as ly119-1 cultured under the T5 treatment were the B group, indicating that a 2.0 to 2.5 mg/L 5-HT application alleviated the 20 cm deep-seeding damage on both lines, while the mitigation effect of a 2.5 mg/L 5-HT application under 20 cm deep-seeding stress varied between two lines.

Moreover, the mitigation effect index (MEI) of 5-HT was calculated based on each trait under the CK(+), CK(-), and Tn (*n* = 1, 2, 3, 4, or 5) treatments, by following the method of Zhao et al. [39] (Appendix A). Then, the comprehensive mitigation effect of 5-HT on deep-seeding tolerance under the Tn treatment (U_Tn_) was evaluated by combining the membership function value [40] of MEI values based on multi-trait determination and accessing their average value. The results showed that the U_Tn_ of two maize materials ranged from 0.251 (ZRX87-1 cultured under the T5 treatment) to 0.673 (ZRX87-1 cultured under the T4 treatment), and their coefficient of variation (CV) was 28.6% (Figure 5B). Thereby, the comprehensive mitigation effect of various 5-HT applications on deep-seeding tolerance under the T1 to T5 treatments was significantly different (*p* < 0.05). Further analysis indicated that the T4 treatment (2.5 mg/L 5-HT application under 20 cm deep-seeding stress) demonstrated the best mitigation effect (average U_T4_ = 0.653) on deep-seeding tolerance in both inbred lines, and there was little difference in the two maize genotypes (Figure 5B,C). In conclusion, 2.5 mg/L exogenous 5-HT can be applied in maize deep-seeding production in arid areas to improve the seedling emergence rate and growth.

### 2.6. Impact of 5-HT on the Expression Levels of Candidate Genes

To further explore the effects of 5-HT stimulation on IAA pathways and lignin biosynthesis in maize seedlings under deep-seeding stress, the relative expression levels of eight genes were analyzed by real-time quantitative PCR (RT-qPCR) in the inbred lines ly119-2 and ZRX87-1 seedlings at seeding depths of 3 cm (CK(+)) and 20 cm (CK(-)) and at a seeding depth of 20 cm treated with 2.5 mg/L 5-HT (T4). The results showed that the relative expressions of four candidate genes involved in IAA biosynthesis and signal transduction, i.e., *Zm00001d049601* (tryptophan decarboxylase 1), *Zm00001d039346* (probable indole-3-acetic acid-amidosynthetase GH3.12), *Zm00001d026530* (indole-3-acetic acid-induced protein ARG7), and *Zm00001d049659* (auxin-induced protein 15A), were significantly upregulated in both the mesocotyl and coleoptile of ly119-2 and ZRX87-1 at the 20 cm sowing depth compared with the 3 cm sowing depth (*p* < 0.05). However, 5-HT was able to further induce the expression of the four genes in their mesocotyl and coleoptile at a 20 cm deep-seeding depth (Figure 6A). By contrast, the four candidate genes responsible for the lignin biosynthesis pathway, i.e., *Zm00001d016471* (trans-cinnamate 4-monooxygenase), *Zm00001d005998* (caffeoyl-CoA O-methyl-transferase 1), *Zm00001d032152* (Cinnamoyl CoA reductase 1), and *Zm00001d053554* (peroxidase), displayed significant downregulation in both the mesocotyl and coleoptile of ly119-2 and ZRX87-1 at the 20 cm sowing depth relative to the normal sowing depth (*p* < 0.05), while 5-HT could further inhibit the expression levels of these genes in their mesocotyl and coleoptile at a 20 cm deep-seeding depth (Figure 6A). In addition, Pearson correlation analysis further showed that the relative expression levels of the eight candidate genes were significantly (*p* < 0.05) correlated with the IAA accumulation and lignin level in both the mesocotyl (Figure 6B) and coleoptile (Figure 6C). The findings showed that deep-seeding stress and exogenous 5-HT positively regulated the expression of genes associated with IAA biosynthesis and signal transduction but negatively regulated those of lignin biosynthesis, suggesting that the IAA and lignin pathways could cooperatively regulate the elongation of the mesocotyl and coleoptile in maize under deep-seeding stress and by 5-HT treatment.

## 3. Discussion

### 3.1. Domestication and Breeding Potential of the Mesocotyl and Coleoptile

The mesocotyl and coleoptile are two embryonic structures, playing important roles in plants’ response to changing environments and seedling establishment. Numerous studies have confirmed that light is a crucial negative regulator of mesocotyl, coleoptile, or hypocotyl elongation [27,41,42,43]. Like rice [15,44], the mesocotyl and coleoptile elongation in maize seedlings was significantly promoted in the darkness and substantially inhibited by light spectral quality in the following order: blue light > red light > white light [41]. Further analysis revealed that the light-induced inhibition of maize mesocotyl and coleoptile elongation was entirely dependent on the phytohormones’ homeostasis between both tissues [41,43]. Similarly, the two members of the annexin family in *Arabidopsis thaliana*, *AtANN1* and *AtANN2,* were involved in phototropism of the etiolated hypocotyl by regulating auxin distribution [45]. Therefore, due to the unique adaption mechanism of the mesocotyl, coleoptile, and hypocotyl of different plants to light stimulation, the shorter mesocotyl was detected in the tropical germplasms compared with the U.S./Canadian corn belt lines [46]. The mesocotyl of indica rice in (sub-) tropical regions was longer than that of japonica rice, which was distributed in the temperate zone [47]. Furthermore, analysis using a wide geographic collection of *Arabidopsis* showed that accessions from latitudes closer to the equator had longer hypocotyls than those from further north [48]. These findings suggested that a loss of light responsiveness at the early seedling stage in various plants is accompanied with selection by breeders and natural domestication in different regions.

In addition, the deep seeding is not conducive to seedlings’ emergence in maize. There was a significantly positive correlation between the cooperative elongation of the mesocotyl and coleoptile and the seedling emergence rate in maize under deep-seeding environments, and its mesocotyl contributed more [2,49,50], supporting our findings in this study. We found an increase in seeding depth from 3 cm (CK(+)) to 20 cm (CK(-)), whereas the seedling emergence rate of both inbred lines (ly119-2 and ZRX87-1) decreased, with their mesocotyls and coleoptiles significantly elongated and slender and weaker seedlings (Figure 1 and Figure 2). The Pearson pairwise correlation analysis further showed that the seedling rate was significantly correlated with the mesocotyl length, coleoptile length, mesocotyl and coleoptile total length, mesocotyl-to-coleoptile ratio, coleoptile coarse, and seedling length (Figure 4B). Furthermore, maize has a long history of domestication [51], and there is an extremely rich abundance of germplasms. The broad-sense heritability of the mesocotyl length (91.0%), coleoptile length (80.9%), mesocotyl and coleoptile total length (91.6%), and mesocotyl-to-coleoptile ratio (86.5%) was high [49], and the differences in mesocotyl and coleoptile characteristics among different maize germplasms were obvious [3]. Thereby, the cooperative elongation of the mesocotyl and coleoptile can serve as a reliable indicator or standard for cultivating elite maize varieties with strong deep-seeding tolerance, which are beneficial for rapid seed germination under adverse environments.

### 3.2. Response Mechanisms of the Mesocotyl and Coleoptile Elongation

Plants have elaborate regulatory mechanisms, facilitating themselves to adapt developmental and environmental cues, including those modulating mesocotyl and coleoptile elongation and stress responses [1,7,41,49,52,53]. Increasing evidence has shown that the mesocotyl and coleoptile elongation, which was attributed to cell division and cell growth (turgor-driven wall expansion), was largely regulated by internal signaling molecules and environmental factors like deep seeding and light.

Among the above internal signaling molecules, phytohormones were vital internal regulators that clearly affected mesocotyl and coleoptile elongation. For instance, exogenous IAA at 10^−6^ to 10^−4^ M applied to roots stimulated maize mesocotyl elongation under deep-seeding treatments, while the auxin transport inhibitors such as triiodobenzoic acid (TIBA) at 10^−5^ to 10^−4^ M applied to roots inhibited the mesocotyl elongation [32]. In this study, a physiological analysis showed that exposure to 20 cm deep-seeding stress significantly increased the IAA level in both the mesocotyl (181.3% and 54.8% in ly119-2 and ZRX87-1, respectively) and coleoptile (14.7% and 10.7% in ly119-2 and ZRX87-1, respectively), compared to the normal sowing depth of 3 cm (Figure 3A,B), which was consistent with previous studies on maize [1,5]. At the same time, the IAA level increased in parallel with the mesocotyl and coleoptile growth under the 20 cm deep-seeding treatment, with the elongation of the mesocotyl and coleoptile yet the weakened mesocotyl coarse and coleoptile coarse (Figure 2A–D). We speculated that the increased IAA accumulation is likely accompanied with cell elongation of the deep-seeding maize mesocotyl and coleoptile, as well as cell wall releasing activity, resulting in a continuous relaxation of the cell wall and mesocotyl/coleoptile elongation through cell growth [32,41]. It has also been reported that *ABP1* (IAA-binding protein) was actively involved in mesocotyl elongation after light stimulation [54]. Moreover, the gene expression profiling showed that the expression levels of four genes (*Zm00001d049601*, *Zm00001d039346*, *Zm00001d026530*, and *Zm00001d049659*) involved in IAA biosynthesis and signal transduction were highly induced in both the mesocotyl and coleoptile of ly119-2 and ZRX87-1 under the 20 cm deep-seeding condition (Figure 6A). RNA sequencing (RNA-Seq) analysis also identified nine genes encoding aldehyde dehydrogenase (TAR2, involved in auxin synthesis), of which four genes (*GRMZM2G122172*, *GRMZM2G071021*, *Zm00001d053904*, and *Zm00001d053906*) were upregulated, whereas another five genes (*GRMZM2G155502*, *GRMZM2G103546*, *GRMZM2G001898*, *Zm00001D051754*, and *Zm00001d038841*) in the mesocotyl of W64A maize seedlings were downregulated at a 20 cm seeding depth [1]. These findings indicated that the mesocotyl and coleoptile elongation in maize is likely controlled by the dynamic changes in IAA caused by the expression of genes responsible for IAA biosynthesis and signal transduction.

It is well known that different soil layers have different micro-environments, including compaction, moisture, temperature, and oxygen. Under the 20 cm deep-seeding condition, reactive oxygen species (ROS), such as H_2_O_2_ production, were increased by 250.7% and 99.1% in the mesocotyl and coleoptile of ly119-2 and ZRX87-1, respectively (Figure 3C,D). On one hand, the plant antioxidant defense systems that scavenge or detoxify overproduced ROS have also been reported to be involved in regulating deep-seeding stress [3]. For instance, using integrated quantitative trait locus (QTL) mapping, meta-analysis, and RNA-Seq, Zhao et al. [49] detected three *POD* (*GRMZM2G040638*, *GRMZM2G107228*, and *GRMZM2G450233*) one catalase (CAT, *GRMZM2G088212*), and one ascorbate peroxidase (APX, *GRMZM2G460406*) candidate gene in four meta-QTL regions controlling maize mesocotyl length, coleoptile length, plumule length, seedling length, and seedling emergence rate under different sowing depth treatments [49]. Chen et al. [55] also mapped a peroxidase (*POD*) gene, i.e., *GRMZM2G035506,* that was located near the PZE-105098349 single-nucleotide polymorphism (SNP) and regulated the deep-seeding maize mesocotyl via genome-wide association analysis (GWAS). In line with these genetic studies, physiological analysis showed that the activities of superoxide dismutase (SOD) and POD were increased and APX and CAT were decreased in four deep-seeding maize mesocotyls and coleoptiles [3].

ROS are also reported to function as important signaling molecules, which indirectly regulate lignin formation and cell wall lignification. For example, polyamine oxidase (PAO) activity was significantly increased in maize mesocotyls cultured in a light environment. Following the oxidative decomposition of polyamines (PAs) and H_2_O_2_ production, the oxidization of lignin monomers (hyproxyphenyl-, guaiacyl-, and syringyl-unit), induced by active POD, occurred, and they were subsequently transformed into lignin, resulting in the hardening of the cell wall and inhibition of mesocotyl elongation [18]. Moreover, the mutants of *polyamine oxidase 5* (*OsPAO5*) in rice synthesized more ET and produced lower amounts of H_2_O_2_, resulting in a longer mesocotyl, faster seedling emergence, and higher yield potential [56]. In parallel, in lignin biosynthesis, exposure to light treatment significantly increased phenylalanine ammonia-lyase (PAL, the first key enzyme in lignin biosynthesis) activity and lignin level in the maize mesocotyl compared to darkness control [41]. In agreement with previous results [1,5], the physiological analysis in this study also showed that lignin contents were significantly decreased by 23.8% and 18.5% in the mesocotyl and coleoptile of two maize lines under 20 cm stress (Figure 3E,F). In addition, the gene expression profiling showed that four lignin biosynthesis genes (*Zm00001d016471*, *Zm00001d005998*, *Zm00001d032152*, and *Zm00001d053554*) were significantly downregulated in both the mesocotyl and coleoptile of ly119-2 and ZRX87-1 at a 20 cm sowing depth compared to the 3 cm control (Figure 6A). The downregulation of lignin biosynthetic genes will likely result in the suppression of some intermediate metabolites involved in lignin biosynthesis.

### 3.3. Mechanisms of Exogenous 5-HT Promoted Mesocotyl and Coleoptile Elongation

In general, IAA, 5-HT, and melatonin (MT) are the most common growth-stimulating indole compounds. A growing body of works has found that IAA promoted maize mesocotyl elongation, induced a steady elevation of H_2_O_2_ content in the apoplast, induced an increase in the NADH-oxidase activity of peroxidases, and induced a transient decrease in oxalate oxidase activity [32,57]. Kutschera and Wang [33] also presented a model to explain growth-limiting proteins in maize coleoptiles and the auxin-brassinosteroid crosstalk hypothesis of mesocotyl elongation. Early studies also showed that MT promoted coleoptile growth in four monocot species including wheat, barley, *Avena sativa* L. (oat), and *Phalaris. Canariensis.* L. (canary grass), by which a relative auxinic activity (with respect to IAA) between 10% and 55% was identified [58]. Until now, however, little has been known about how 5-HT controls the elongation of the mesocotyl and coleoptile of 20 cm deep-seeded maize seedlings.

To address this question, we investigated the effects of five concentrations of exogenous 5-HT on the seedling emergence and growth, cooperative elongation of both the mesocotyl and coleoptile, as well as the mesocotyl and coleoptile thickening of two maize seedlings at the deep-seeding depth of 20 cm (Figure 1 and Figure 2). It clearly showed that while different concentration exogenous 5-HT mostly stimulated the elongation of the mesocotyl and coleoptile at a 20 cm sowing depth, the endogenous IAA level was increased, and H_2_O_2_ production and lignin content were decreased differentially (Figure 3). These results indicated the co-existence of IAA and 5-HT in both the mesocotyl and coleoptile of maize seedlings, where they might co-participate in some physiological processes as auxin hormones in maize. In these regards, according to the values of all 17 traits under all treatments, we calculated the MEI values of 5-HT under 20 cm deep-seeding stress (Appendix A) and examined the comprehensive mitigation effect of 5-HT under 20 cm deep-seeding stress (U_Tn_) via the membership function method. The results showed that the optimum concentration of 5-HT for relieving deep-seeding stress was 2.5 mg/L (Figure 5B,C), providing a guidance for practical application of 5-HT to improve the seedlings’ emergence and yield potential under deep-seeding conditions in arid areas.

Altogether, mesocotyl and coleoptile development are two crucial agronomic traits for deep-seeding maize production regarding their potential role in seedlings’ establishment. In this study, according to the phenotypic variations (Figure 1 and Figure 2), physiological characteristics (Figure 3), expression patterns of IAA- and lignin-related genes (Figure 6A), and the pairwise Pearson correlations (Figure 4B and Figure 6B,C) of the mesocotyl and coleoptile in two maize lines under seven treatments, we proposed a model for a possible molecular network regulating the exogenous 5-HT-induced cooperative elongation of mesocotyls and coleoptiles in deep-seeded maize seedlings (Figure 7). Briefly, on one hand, when maize seeds are cultured at a 20 cm sowing depth, exogenous 5-HT (2.5 mg/L) application is able to activate the expression of the IAA biosynthesis and signal transduction genes, whereas it suppresses the expression of lignin biosynthesis genes. On the other hand, the activation of essential genes in these pathways by 5-HT is coordinated in a highly interconnected network to activate or inhibit the secondary and tertiary metabolic products, such as the dynamic changes in endogenous IAA, H_2_O_2_, and lignin levels. The enhanced IAA contents yet decreased lignin and H_2_O_2_ levels positively stimulate the longitudinal and lateral elongation of mesocotyl and coleoptile cells, resulting in mesocotyl and coleoptile elongation and thickening, seedling emergence, and growth at the deep-seeding soil layer. The work may lay a foundation for understanding maize’s deep-seeding tolerance mechanism, and 5-HT at 2.5 mg/L can be used as a simple and practicable strategy to improve maize’s deep-seeding tolerance during seed germination.

## 4. Materials and Methods

### 4.1. Maize Materials and Experimental Design

The two deep-seeding-intolerant maize inbred lines ly119-2 and ZRX87-1 from Longxi experimental stations, Gansu, China (34.97° N, 104.40° E, 2074 m altitude), were used in this study. Uniform seeds were first sterilized with 70% (*v*/*v*) ethanol solution for 10 min and then rinsed five times with double-distilled water (ddH_2_O). The sterilized seeds were soaked in darkness for 48 h in six concentrations of exogenous 5-HT (Sigma-Aldrich Ltd., Shanghai, China; CAS: 50-67-9) solution (i.e., 0, 1.0, 1.5, 2.0, 2.5, and 3.0 mg/L). Each of the above 5-HT solutions was mixed separately with sterilized vermiculite uniformly in a proportion of 100 mL: 500 g to prepare the seed sowing matrix. The seed sowing matrix of each 5-HT concentration was loaded to a certain height in the seed deep-seeding test device (Chinese Patent; CN 209768182U). The 30 seeds of each inbred line soaked with the corresponding concentration of 5-HT were then evenly seeded on the seeding matrix. Finally, these seeds were covered with the corresponding seed sowing matrix to maintain depths of 3 cm and 20 cm. Totally, an odd number of treatments were designed for this experiment, including 3 cm seeding depth + 0 mg/L 5-HT application (CK(+)), 20 cm seeding depth + 0 mg/L 5-HT application (CK(-)), 20 cm seeding depth + 1.0 mg/L 5-HT application (T1), 20 cm seeding depth + 1.5 mg/L 5-HT application (T2), 20 cm seeding depth + 2.0 mg/L 5-HT application (T3), 20 cm seeding depth + 2.5 mg/L 5-HT application (T4), and 20 cm seeding depth + 3.0 mg/L 5-HT application (T5). Each treatment was repeated five times. All deep-seeding test devices were then transferred to an intelligent artificial climate room (the culture conditions were set as 12/12 h light/dark cycle, 20 ± 0.5 °C constant temperature, 300 μM m^-2^ S^-1^ light intensity, and 60% relative humidity) for ten days. Each deep-seeding test device was replenished with 20 mL of corresponding 5-HT solution every two days. 

### 4.2. Phenotypic and Physiological Assays

After ten days of seed germination under the above seven treatments, the seedling emergence rate was counted. Then, the seedlings were quickly washed off with ddH_2_O, and five germinated seedlings with the same overall growth were selected to measure growth parameters, including the mesocotyl length, coleoptile length, mesocotyl coarse, coleoptile coarse, mesocotyl and coleoptile total length, mesocotyl-to-coleoptile ratio, mesocotyl weight, coleoptile weight, seedling length, and seedling weight [41,49].

The separated mesocotyls and coleoptiles under each treatment were frozen in liquid nitrogen immediately and stored at −80 °C for physiological assays. Liquid chromatography-tandem mass spectrometry (UHPLC-MS/MS) analysis was used to measure the IAA content in mesocotyls and coleoptiles [41]. According to Zhao et al.’s [39] method, H_2_O_2_ production in the mesocotyl and coleoptile was analyzed at 410 nm via an ultraviolet spectrophotometer. The optical absorption value of the corresponding samples at 280 nm [41] was determined by ultraviolet spectrophotometry to estimate the lignin content in the mesocotyl and coleoptile.

### 4.3. Mitigation Effect Index (MEI) Value and Comprehensive Evaluation Analysis

To objectively evaluate the mitigation effect of different 5-HT applications on a single deep-seeding-tolerant trait under 20 cm deep-seeding stress, the MEI value [39] was calculated as follows:*MEI_i-Tn_* = [*T_i-Tn_* − *T_i-CK(-)_*]/|*T_i-CK(+)_* − *T_i-CK(-)_*|(1)
*MEI_i-Tn_* = [*T_i-CK(-)_* − *T_i-Tn_*]/|*T_i-CK(+)_* −* T_i-CK(-)_*|(2)
where *MEI_i__-Tn_* is the mitigation effect index of the corresponding 5-HT concentration application on the *i*-th trait under the Tn treatment (*n* = 1, 2, 3, 4, and 5). *T_i__-CK(+)_* is the *i*-th trait value under the CK(+) treatment, *T_i__-CK(-)_* is the *i*-th trait value under the CK(-) treatment, and *T_i__-Tn_* is the *i*-th trait value under the Tn treatment. When the *i*-th trait was positively or negatively correlated with maize deep-seeding tolerance, Equation (1) or Equation (2) was estimated, respectively. Then, the membership function value [40] was used to comprehensively evaluate the mitigation effect of 5-HT on deep-seeding stress as follows: *U_i-Tn_* = [*MEI_i-Tn_* − *MEI_i-min_*]/[*MEI_i-max_* − *MEI_i-min_*](3)
*U_i-Tn_* = 1 − [*MEI_i-Tn_* − *MEI_i-min_*]/[*MEI_i-max_* − *MEI_i-min_*](4)
(5)$$U_{Tn}=\frac{1}{17}\overset{17}{\underset{i=1}{\sum }}U_{i-Tn}$$
where *U_i__-Tn_* is the membership function value of the MEI for the *i*-th trait under the Tn treatment, *MEI_i__-max_* is the maximum value of the MEI for the *i*-th trait under all treatments, and *MEI_i__-min_* is the minimum value of the MEI for the *i*-th trait under all treatments. When the *i*-th trait was positively or negatively correlated with maize deep-seeding tolerance, Equation (3) or Equation (4) was calculated, respectively. *U_Tn_* is the average value of the membership function of the mitigation effect of 5-HT on deep-seeding stress under the Tn treatment.

### 4.4. Statistical Analysis

For all tested traits of the two maize inbred lines under all treatments, we used IBM-SPSS Statistics v19.0 (SPSS Inc., Chicago, IL, USA, https://www.Ibm.com/products/spss-statistics, accessed on 20 June 2023) to perform analysis of variance (ANOVA) and principal component analysis (PCA). The between-group linkage cluster analysis was performed using R package (R version 4.2.3; http://www.R-project.org/, accessed on 3 September 2023). The Pearson correlation coefficient diagram was made with the Genescloud tool (https://www.genescloud.cn, accessed on 3 September 2023).

### 4.5. Total RNA Extraction and Quantitative Real-Time PCR (qRT-PCR) Analysis

Because the best mitigation effect of deep-seeding tolerance on maize seedlings was that under the T4 treatment (Figure 5B,C), the mesocotyl and coleoptile of both ly119-2 and ZRX87-1 seedlings under the CK(+), CK(-), and T4 treatments were used for gene expression analysis. Their total RNA was extracted with TRIZOL reagent (TIANGEN, Beijing, China), and 0.5 μg RNA was then reverse-transcribed to produce first-stand cDNA using HiScript^®^Q RT SuperMix, Vazyme, Nanjing, China, according to the manufacturer’s protocol. The qRT-PCR was conducted using TransStart TIP Green qPCRSuperMix (Tran, Beijing, China). Primer Premier 5.0 online software (http://www.premierbiosoft.com/) was used to design the primers for eight candidate genes (Appendix A). AgBase v2.00 (https://agbase.arizona.edu/) [59] was used to annotate candidate genes’ functions. The relative gene expression level was assessed by the 2^−△Ct^ method, with *Zm00001d010159* (*Actin 1*) as an internal reference gene [60].

## Figures and Tables

**Figure 1 ijms-24-17061-f001:**
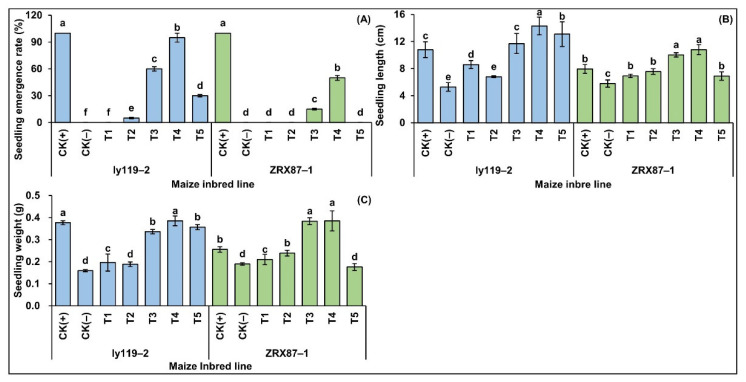
Maize seedling phenotypic variations under different treatments, including 3 cm seeding depth + 0 mg/L serotonin (5-HT) application (CK(+)), 20 cm seeding depth + 0 mg/L 5-HT application (CK(-)), 20 cm seeding depth + 1.0 mg/L 5-HT application (T1), 20 cm seeding depth + 1.5 mg/L 5-HT application (T2), 20 cm seeding depth + 2.0 mg/L 5-HT application (T3), 20 cm seeding depth + 2.5 mg/L 5-HT application (T4), and 20 cm seeding depth + 3.0 mg/L 5-HT application (T5). (**A**) Seedling emergence rate. (**B**) Seedling length. (**C**) Seedling weight. Different lowercase letters among treatments within single maize line indicate significant differences at *p* < 0.05.

**Figure 2 ijms-24-17061-f002:**
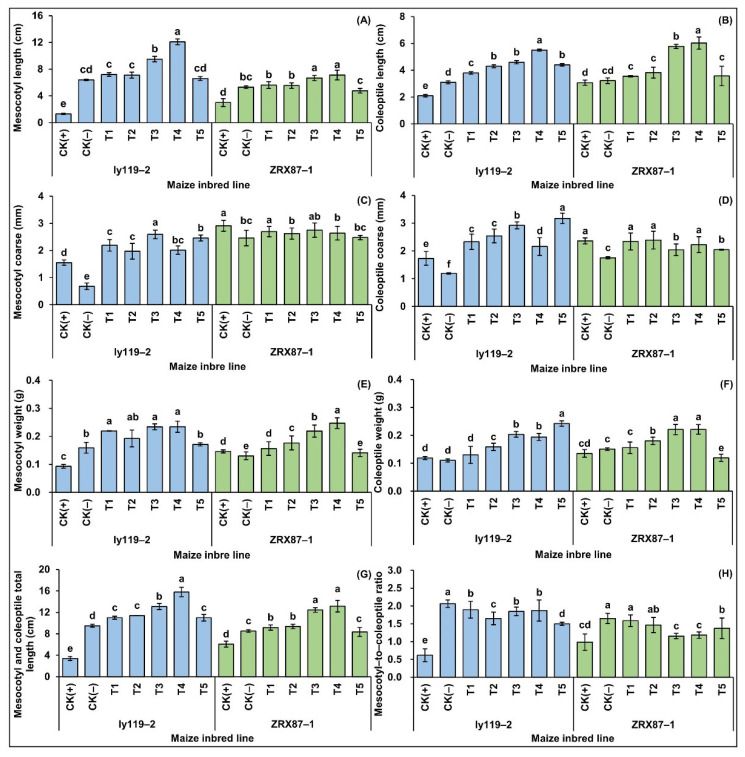
Maize growth phenotypes of mesocotyl and coleoptile under different treatments, including 3 cm seeding depth + 0 mg/L serotonin (5-HT) application (CK(+)), 20 cm seeding depth + 0 mg/L 5-HT application (CK(-)), 20 cm seeding depth + 1.0 mg/L 5-HT application (T1), 20 cm seeding depth + 1.5 mg/L 5-HT application (T2), 20 cm seeding depth + 2.0 mg/L 5-HT application [T3], 20 cm seeding depth + 2.5 mg/L 5-HT application (T4), and 20 cm seeding depth + 3.0 mg/L 5-HT application (T5). (**A**) Mesocotyl length. (**B**) Coleoptile length. (**C**) Mesocotyl coarse. (**D**) Coleoptile coarse. (**E**) Mesocotyl weight. (**F**) Coleoptile weight. (**G**) Mesocotyl and coleoptile total length. (**H**) Mesocotyl-to-coleoptile ratio. Different lowercase letters among treatments within single maize line indicate significant differences at *p* < 0.05.

**Figure 3 ijms-24-17061-f003:**
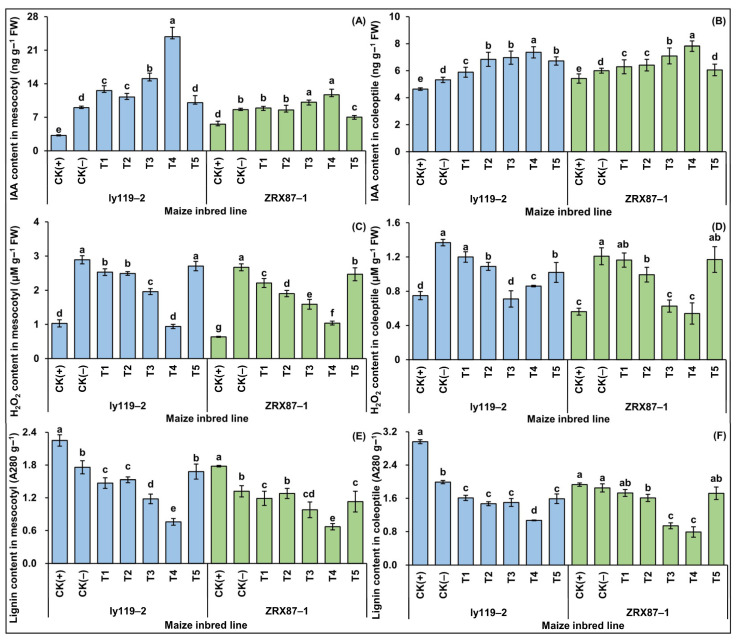
Physiological responses of maize mesocotyl and coleoptile under different treatments, including 3 cm seeding depth + 0 mg/L serotonin (5-HT) application (CK(+)), 20 cm seeding depth + 0 mg/L 5-HT application (CK(-)), 20 cm seeding depth + 1.0 mg/L 5-HT application (T1), 20 cm seeding depth + 1.5 mg/L 5-HT application (T2), 20 cm seeding depth + 2.0 mg/L 5-HT application (T3), 20 cm seeding depth + 2.5 mg/L 5-HT application (T4), and 20 cm seeding depth + 3.0 mg/L 5-HT application (T5). (**A**) Indole-3-acetic acid (IAA) in mesocotyl. (**B**) IAA in coleoptile. (**C**) Hydrogen peroxide (H_2_O_2_) in mesocotyl. (**D**) H_2_O_2_ in coleoptile. (**E**) Lignin content in mesocotyl. (**F**) Lignin content in coleoptile. Different lowercase letters among treatments within the mesocotyl or coleoptile of a single maize line indicate significant differences at *p* < 0.05.

**Figure 4 ijms-24-17061-f004:**
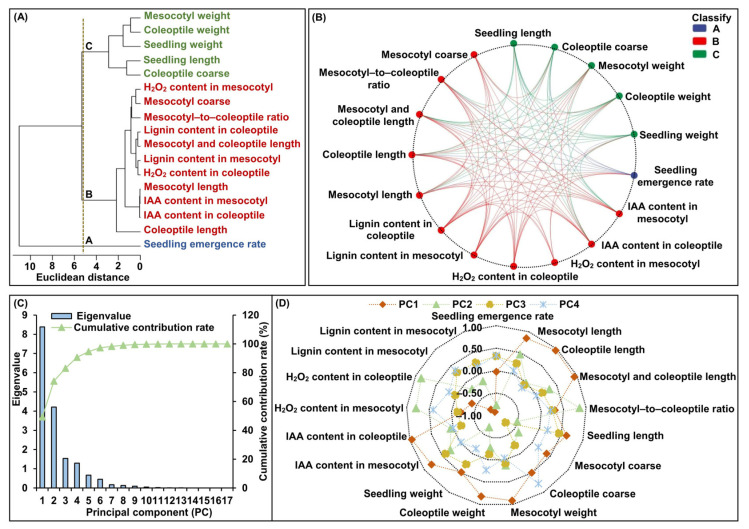
Framework of relations based on between-group linkage cluster analysis, Pearson pairwise correlation analysis, and principal component analysis (PCA) of 17 tested traits of two maize inbred lines treated with or without serotonin (5-HT) application at 20 cm and 3 cm seeding depths. (**A**) Between-group linkage cluster analysis was performed using R package (http://www.R-project.org/, accessed on 3 September 2023). (**B**) The Pearson correlation coefficient diagram among 17 tested traits was made with the Genescloud tool (https://www.genescloud.cn, accessed on 3 September 2023). Lines represent significant correlations (*p* < 0.05). (**C**) Eigenvalues and cumulative variance of principal components (PCs). (**D**) Eigenvector distributions of 17 tested traits in PC1, PC2, PC3, and PC4. H_2_O_2_: hydrogen peroxide; IAA: indole-3-acetic acid.

**Figure 5 ijms-24-17061-f005:**
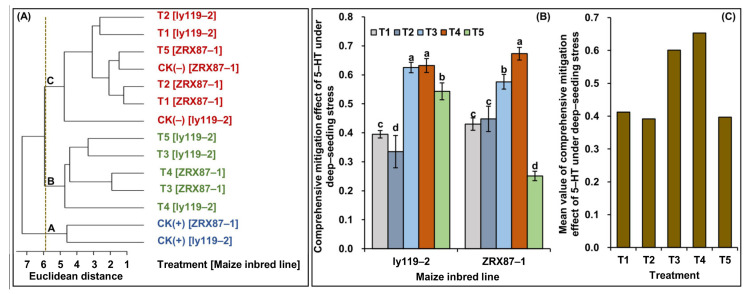
Comprehensive evaluations of deep-seeding tolerance or mitigation effect of serotonin (5-HT) on deep-seeding tolerance under different treatments in two maize inbred lines (ly119-2 and ZRX87-1). CK(+): 3 cm seeding depth + 0 mg/L 5-HT application; CK(-): 20 cm seeding depth + 0 mg/L 5-HT application; T1: 20 cm seeding depth + 1.0 mg/L 5-HT application; T2: 20 cm seeding depth + 1.5 mg/L 5-HT application; T3: 20 cm seeding depth + 2.0 mg/L 5-HT application; T4: 20 cm seeding depth + 2.5 mg/L 5-HT application; T5: 20 cm seeding depth + 3.0 mg/L 5-HT application. (**A**) Between-group linkage cluster analysis was performed using R package (http://www.R-project.org/, accessed on 3 September 2023). (**B**) The comprehensive mitigation effects of 5-HT on deep-seeding tolerance under different treatments in two inbred lines. Different lowercase letters among treatments within a single maize line indicate significant differences at *p* < 0.05. (**C**) Mean value of comprehensive mitigation effects of 5-HT on deep-seeding tolerance under different treatments in both maize materials.

**Figure 6 ijms-24-17061-f006:**
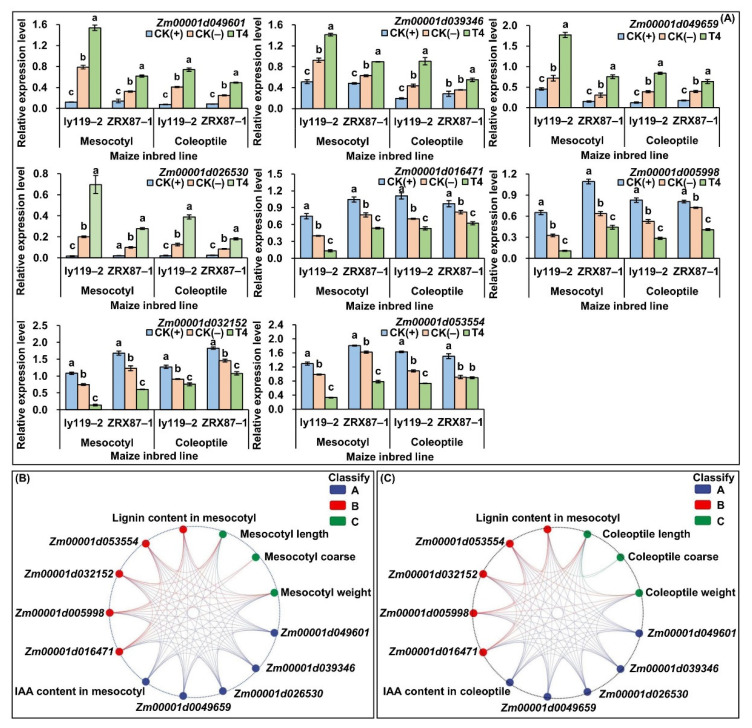
The relative expression patterns of eight candidate genes involved in indole-3-acetic acid (IAA) biosynthesis and signal transduction and lignin biosynthesis pathways, growth phenotypes, physiological responses (IAA and lignin contents), and their Pearson correlations in the mesocotyl and coleoptile of two maize inbred lines (ly119-2 and ZRX87-1) under 3 cm seeding depth + 0 mg/L serotonin (5-HT) application (CK(+)), 20 cm seeding depth + 0 mg/L 5-HT application (CK(-)), and 20 cm seeding depth + 2.5 mg/L 5-HT application (T4) treatments. (**A**) The relative expression levels of the eight candidate genes. Different lowercase letters among treatments within the mesocotyl of coleoptile of a single maize line indicate significant differences at *p* < 0.05. (**B**) The correlational relationships in the mesocotyl among growth phenotypes (mesocotyl length, coarse, and weight), physiological responses (IAA and lignin contents), and eight candidate genes were analyzed by the Genescloud tool (https://www.genescloud.cn, accessed on 13 October 2023). Lines represent significant correlations (*p* < 0.05). (**C**) The correlational relationships in the coleoptile among growth phenotypes (coleoptile length, coarse, and weight), physiological responses (IAA and lignin contents), and eight candidate genes were analyzed by the Genescloud tool (https://www.genescloud.cn, accessed on 13 October 2023). Lines represent significant correlations (*p* < 0.05).

**Figure 7 ijms-24-17061-f007:**
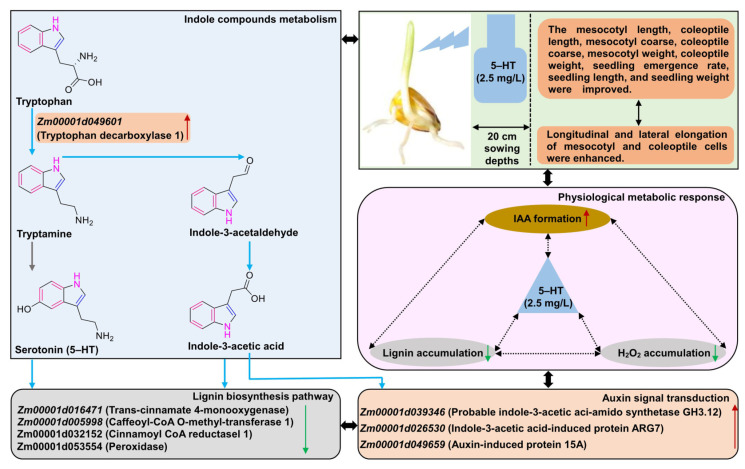
Molecular network underlying the cooperative elongation of mesocotyl and coleoptile in response to exogenous serotonin (5-HT) stimulation and deep-seeding stress. H_2_O_2_: hydrogen peroxide; IAA: indole-3-acetic acid. The blue arrow indicates that the path proceeds; the gray arrow indicates that the path is blocked; the red arrow indicates that the genes/metabolites are upregulated/accumulated; the green arrow indicates that the genes/metabolites are downregulated/accumulated.

## Data Availability

Data is contained within the article and Appendix A.

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
