# Peer review of "Exogenous Serotonin (5-HT) Promotes Mesocotyl and Coleoptile Elongation in Maize Seedlings under Deep-Seeding Stress through Enhancing Auxin Accumulation and Inhibiting Lignin Formation"

_ijms, 2023, doi:10.3390/ijms242317061_

Round 1

Reviewer 1 Report

Comments and Suggestions for Authors

The publication describes the effect of serotonin on IAA metabolism and inhibition of lignin formation in corn seedlings planted under deep-seeding stress. This publication seems to be within the scope of journal. However it needs several corrections to be more acceptable for publication.

1.      I suggest changing in the title of the work, in the abstract and in the keywords 5-hydroxytryptamine to serotonin, because this compound is better known under this second name. In my opinion, this small change may ensure better positioning of the publication in the future. In line 18: it should be „Serotonin (5-hydroxytryptamine, 5-HT)”.

2.      Lines 75-80 should include information that indole derivatives are auxins, because IAA is natural auxin.

3.      In publication: Duca, D.R., Glick, B.R. Indole-3-acetic acid biosynthesis and its regulation in plant-associated bacteria. Appl Microbiol Biotechnol 104, 8607–8619 (2020), the IAA biosynthesis is well described. I suggest changing the name from 2-(1H-indol-3-yl)acetalaldehyde to indole-3-acetaldehyde. It is also possible to add the name of other enzymes based on this publication.

4.      It is not clear why the discussion mentions the influence of different wavelengths of light on the growth of mesocotyl and coleoptile, since this was not the subject of the authors' research. I think this fragment can be removed without loss of publication quality.

Author Response

Reviewer 1:

The publication describes the effect of serotonin on IAA metabolism and inhibition of lignin formation in corn seedlings planted under deep-seeding stress. This publication seems to be within the scope of journal. However it needs several corrections to be more acceptable for publication.

Thanks for your positive comments. As suggested, we have carefully revised and improved the manuscript. We then have re-submitted the manuscript.

Thank you for your consideration.

  1. I suggest changing in the title of the work, in the abstract and in the keywords 5-hydroxytryptamine to serotonin, because this compound is better known under this second name. In my opinion, this small change may ensure better positioning of the publication in the future.In line 18: it should be „Serotonin (5-hydroxytryptamine, 5-HT)”.

Thanks for your positive comments. As suggested, we have changed the word in the manuscript and Supplementary materials, “5-hydroxytryptamine” to “serotonin”.  We then have re-submitted the manuscript.

Thank you for your consideration.

  1. Lines 75-80 should include information that indole derivatives are auxins, because IAA is natural auxin.

Thanks for your positive comments. As suggested, we have added the corresponding contents in Line 75 of the manuscript, and then have re-submitted the manuscript.

Thank you for your consideration.

  1. In publication: Duca, D.R., Glick, B.R. Indole-3-acetic acid biosynthesis and its regulation in plant-associated bacteria. Appl Microbiol Biotechnol 104, 8607–8619 (2020), the IAA biosynthesis is well described. I suggest changing the name from 2-(1H-indol-3-yl) acetalaldehyde to indole-3-acetaldehyde. It is also possible to add the name of other enzymes based on this publication.

Thanks for your positive comments. As suggested, we have changed the word in the Figure 7, “2-(1H-indole-3-yl) acetalaldehyde” to “indole-3-acetaldehyde”, “5-hydroxytryptamine” to “serotonin”. We have provided a new Figure 7, and then have re-submitted the manuscript.

Thank you for your consideration.

  1. It is not clear why the discussion mentions the influence of different wavelengths of light on the growth of mesocotyl and coleoptile, since this was not the subject of the authors' research. I think this fragment can be removed without loss of publication quality.

Thanks for your positive comments. It is well known that different light signal can significant influence the elongation of mesocotyl, coleoptile, or hypocotyl in multiple plants including maize (Zhao, X.Q.; Niu, Y.N.; Hossain, Z.; Zhao, B.Y.; Bai, X.D.; Mao , T.T. New insights into light spectral quality inhibits the plasticity elongation of maize mesocotyl and coleoptile during seed germination. Front. Plant Sci. 2023, 14, 1152399), rice (Feng, F.J.; Mei, H.W.; Fan, P.Q.; Li, Y.N.; Xu, X.Y.; Wei, H.B.; Yan, M.; Luo, L.J. Dynamic transcriptome and phytohormone profiling along the time of light exposure in the mesocotyl of rice seedling. Sci. Rep. 2017, 7, 11961), and Arabidopsis thaliana (Hao, Y.H.; Zeng, Z.X.; Zhang, X.L.; Xie, D.X.; Li, X.; Ma, L.B.; Liu, M.Q.; Liu, H.T. Green means go: green light promotes hypocotyl elongation via brassinosteroid signaling. Plant Cell, 2023, koad022).

In the fact, deep-sowing of maize seeds, which was in the darkness, to receive light for autotrophic growth, germinated seeds need to quickly penetrate the soil surface, the mesocotyl could pushes the coleoptile across the soil surface in maize during germination, whereas the mesocotyl was inhibited by light as soon as the coleoptile sprouts from the soil surface (Zhao, x.q.; Zhong, Y. 24-Epibrassinolide mediated interaction among antioxidant defense, lignin metabolism, and phytohormones signaling and coleoptile under deep-seeding stress. Russ. J. Plant Physiol. 2021, 68, 1194-1207). Therefore, deep-sowing of seeds is affected by soil resistance, light signal, temperature, and so on. In above regards, we discussed the influence of different wavelengths of light on the growth of mesocotyl and coleoptile in the Discussion mentions, the discussion can better reveal the effect of deep-sowing on mesocotyl and coleoptile elongation in maize.

Thank you for your consideration.

Open Review: I would not like to sign my review report.

Thanks for your positive comments.

Thank you for your consideration.

Quality of English Language: I am not qualified to assess the quality of English in this paper.

Thanks for your positive comments.

Thank you for your consideration.

Does the introduction provide sufficient background and include all relevant references? Yes.

Thanks for your positive comments.

Thank you for your consideration.

Are all the cited references relevant to the research? Yes.

Thanks for your positive comments.

Thank you for your consideration.

Is the research design appropriate? Yes.

Thanks for your positive comments.

Thank you for your consideration.

Are the methods adequately descrided? Yes.

Thanks for your positive comments.

Thank you for your consideration.

Are the results clearly presented? Yes.

Thanks for your positive comments.

Thank you for your consideration.

Are the conclusions supported by the results? Yes.

Thanks for your positive comments.

Thank you for your consideration.

Reviewer 2 Report

Comments and Suggestions for Authors

The manuscript “Exogenous 5–Hydroxytryptamine (5–HT) Promotes Mesocotyl and Coleoptile Elongation of Maize Seedling under Deep–seeding Stress through Enhancing Auxin Accumulation and Inhibiting Lignin Formation” is a very interesting study. Just a few recommendations to the authors:

 1-      In Abstract section, the authors are recommended to highlight with more emphasis the results obtained.

2-      In Introduction section, it is recommended to the authors to add a reference for the line 46, the added web page is too wide.

3-      The authors carried out a study using vermiculite as a substrate, but what would happen with the use of another type of substrate that exerts greater pressure for the emergence of the seedling?

4-      Could the authors explain why they did not include evaluations with the addition of 5-HT at a depth of 3 cm?

5-      Authors are recommended to review the manuscript, for phrases as "20cm" that is widespread in all sections, “viabetween”, “wet investigated”, etc.

Author Response

Reviewer 2:

The manuscript “Exogenous 5–Hydroxytryptamine (5–HT) Promotes Mesocotyl and Coleoptile Elongation of Maize Seedling under Deep–seeding Stress through Enhancing Auxin Accumulation and Inhibiting Lignin Formation” is a very interesting study. Just a few recommendations to the authors:

Thanks for your positive comments. As suggested, we have carefully revised and improved the manuscript. We then have re-submitted the manuscript.

Thank you for your consideration.

 1- In Abstract section, the authors are recommended to highlight with more emphasis the results obtained.

Thanks for your positive comments. As suggested, we have added the corresponding data in the Abstract section to highlight with more emphasis the results obtained, namely : “Serotonin (5–HT), an indoleamine compound, has been known to mediate many physiological responses of plants under environmental stress. Deepseeding (≥ 20 cm) of maize seeds is an important cultivation strategy to ensure seedling emergence and survival under drought stress. However, the role of 5–HT in maize deep–seeding tolerance remains unexplored. Understanding the mechanisms and evaluating the optimal concentration of 5–HT in alleviating deep-seeding stress could benefit maize production. In this study, two maize inbred lines were treated with or without 5–HT at both sowing depths of 20 cm and 3 cm, respectively. Effects of different concentrations of 5–HT on growth phenotypes, physiological metabolism and genes expression of two maize inbred lines were examined at the sowing depths of 20 cm and 3 cm. Compared to normal seedling depth of 3 cm, the elongation of mesocotyl (average elongated 3.70 cm) and coleoptile (average elongated 0.58 cm), and secretion of indole–3–acetic acid (IAA; average increased 3.73 and 0.63 ng g-1 FW), and hydrogen peroxide (H2O2; average increased 1.95 and 0.63 μM g-1 FW) in mesocotyl and coleoptile were increased under 20 cm stress, with concomitant decrease in ligninsynthesis (average decreased 0.48 and 0.53 A280 g-1). Under 20 cm deepseeding stress, the addition of 5HT activated the expression of multiple genes of IAA biosynthesis and signal transduction, including Zm00001d049601, Zm00001d039346, Zm00001d026530, Zm00001d049659, and it also stimulated IAA production in both mesocotyl and coleoptile of maize seedlings. On the contrary, 5HT suppressed the expression of genes for lignin biosynthesis (Zm00001d016471, Zm00001d005998, Zm00001d032152, and Zm00001d053554), and retarded the accumulation of H2O2 and lignin, resulting in the elongation of the mesocotyl and coleoptile of maize seedlings. Comprehensive evaluation analysis showed that the optimum concentration of 5HT in relieving deepseeding stress was 2.5 mg/L for both inbred lines, 5HT therefore could improve seedling emergence rate and alleviate deepseeding stress on maize seedlings. These findings could provide a novel strategy for improving maize deepseeding tolerance, thus to enhance yield potential under drought and water stress.” We then have re-submitted the manuscript.

Thank you for your consideration.

2- In Introduction section, it is recommended to the authors to add a reference for the line 46, the added web page is too wide.

Thanks for your positive comments. As suggested, we have added corresponding reference ([38] Erland, L.A.E.; Shukla, M.R.; Singh, A.S.; Murch, S.J.; Saxena, P.K. Melatonin and serotonin: Mediators in the symphony of plant morphogenesis. J. Pineal Res. 2018, 64, 1.). We then have re-submitted the manuscript.

Thank you for your consideration.

3- The authors carried out a study using vermiculite as a substrate, but what would happen with the use of another type of substrate that exerts greater pressure for the emergence of the seedling?

Thanks for your positive comments. This is a very good question. In general, vermiculite (Zhao, X.Q.; Niu, Y.N.; Hossain, Z.; Shi, J.; Mao, T.T.; Bai, X.D. Integrated QTL mapping, meta-analysis, and RNA-sequencing reveal candidate genes for maize deep-sowing tolerance. Int. J. Mol. Sci. 2023, 24, 6770. Zhao, X.Q.; Niu, Y.N. The combination of conventional QTL analysis, bulked-segregant analysis, and RNA-Sequencing provide new genetic insights into maize mesocotyl elongation under multiple deep-seeding environments. Int. J. Mol. Sci. 2022, 23, 4223.) and sand (Liu, H.J.; Zhang, L.; Wang, J.C.; Li, C.S.; Zeng, Y.Z.; Liu, S.S.; Hu, S.L.; Wang, J.H.; Lee, M.; Lubberstedt, T.; Zhao, G.W. Quantitative trait locus analysis for deep-sowing germination ability in the maize IBM Syn10 DH population. Front. Plant Sci. 2017, 8, 813. Zhang, H.W.; Ma, P.; Zhao, Z.N.; Zhao, G.W.; Tian, B.H.; Wang, J.H.; Wang, G.Y. Mapping QTL controlling maize deep-seeding tolerance-related traits and confirmation of a major QTL for mesocotyl length. Theor. Appl. Genet. 2012, 124, 223-232.), which were used as substrate for studying maize deep-sowing tolerance. It is well known that sand exerts greater pressure for the emergence of the seedling than that vermiculite. Therefore, in the future, we need to further study the effect of different concentrations of serotonin (5-HT) on growth phenotypes, physiological metabolism, and genes expressions of maize seedlings at 3 cm and 20 cm sand substrates.

Thank you for your consideration.

4- Could the authors explain why they did not include evaluations with the addition of 5-HT at a depth of 3 cm?

Thanks for your positive comments. Like indole-3-acetic acid (IAA), melatonin (MT), serotonin (5-HT) is an important indole derivatives, the maize seeds were treated with or without 5-HT at 3 cm sowing depths, that may affect the seedling emergence, mesocotyl and coleoptile development of maize.

It is well known that “During seed germination, maize is often exposed to drought stress causing growth retardation and significant yield losses worldwide [1–4]. The normal maize sowing depth is 3~5 cm, because of abundant evaporation and moisture loss in soil surface, maize can’t absorb enough water to initiate seedling emergence in shallow sowing [1]. In agricultural practice, maize deepseeding (≥ 20 cm) is an important cultivation strategy to ensure seedlings emergence and survival under drought stress [5–7]. So far, a limited number of maize varieties with strong deepseeding tolerance have been developed and widely used in (semi) arid areas. ” in Lines 46-53 of the manuscript. Thus, it is of no productive value to study the effect of different concentrations of exogenous 5-HT on maize seedling emergence, as well as mesocotyl and coleoptile growth at 3cm sowing depth.

The purpose of this work was “in this study, effect of different concentrations of 5HT on mesocotyl and coleoptile elongation under deep-seeding stress was evaluated using two maize inbred lines. The changes of growth indexes and physiological metabolism, as well as gene expression levels involved in IAA biosynthesis, auxin signal transduction, and lignin biosynthesis, were studied to elucidate the mechanism of 5HT mediated mitigation of deepseeding stress. This study will provide valuable information for alleviating maize deepseeding stress and improving yield, as well as maize drought tolerance breeding.” in Lines 92-99 of the manuscript.

Thank you for your consideration.

5- Authors are recommended to review the manuscript, for phrases as "20cm" that is widespread in all sections, “viabetween”, “wet investigated”, etc.

Thanks for your positive comments. As suggested, we have carefully revised and improved the corresponding contents in the manuscript, namely: “To comprehensively evaluate the mitigation effect of different 5HT application on deep–seeding tolerance, the 17 tested traits of both maize inbred lines under all treatments were used for the qualitative identification of deep–seeding tolerance via between–groups linkage cluster analysis.” and “To address this question, we investigated the effects of five concentrations of exogenous 5–HT on the seedling emergence and growth, cooperative elongation of both mesocotyl and coleoptile, as well as mesocotyl and coleoptile thickening of two maize seedlings at the deep–seeding depth of 20 cm (Figure 1; Figure 2).”. We then have re-submitted the manuscript.

Thank you for your consideration.

Open Review: I would like to sign my review report.

Thanks for your positive comments.

Thank you for your consideration.

Quality of English Language: I am not qualified to assess the quality of English in this paper.

Thanks for your positive comments.

Thank you for your consideration.

Does the introduction provide sufficient background and include all relevant references? Yes.

Thanks for your positive comments.

Thank you for your consideration.

Are all the cited references relevant to the research? Yes.

Thanks for your positive comments.

Thank you for your consideration.

Is the research design appropriate? Can be improved.

Thanks for your positive comments.

Thank you for your consideration.

Are the methods adequately descrided? Yes.

Thanks for your positive comments.

Thank you for your consideration.

Are the results clearly presented? Yes.

Thanks for your positive comments.

Thank you for your consideration.

Are the conclusions supported by the results? Yes.

Thanks for your positive comments.

Thank you for your consideration.
